# Food Classification and Meal Intake Amount Estimation through Deep Learning

**Ji-hwan Kim [1], Dong-seok Lee [2] and Soon-kak Kwon [1],***

[1] Department of Computer Software Engineering, Dong-Eui University, Busan 47340, Republic of Korea; ghksdl1224@gmail.com

[2] AI Grand ICT Center, Dong-Eui University, Busan 47340, Republic of Korea; ulsan333@gmail.com

\* Correspondence: skkwon@deu.ac.kr; Tel.: +82-51-890-1727

**Abstract:** This paper proposes a method to classify food types and to estimate meal intake amounts in pre- and post-meal images through a deep learning object detection network. The food types and the food regions are detected through Mask R-CNN. In order to make both pre- and post-meal images to a same capturing environment, the post-meal image is corrected through a homography transformation based on the meal plate regions in both images. The 3D shape of the food is determined as one of a spherical cap, a cone, and a cuboid depending on the food type. The meal intake amount is estimated as food volume differences between the pre-meal and post-meal images. As results of the simulation, the food classification accuracy and the food region detection accuracy are up to 97.57% and 93.6%, respectively.

**Keywords:** object detection; food classification; food volume estimation; meal intake amount estimation





## 1. Introduction

For people who need strict dietary management such as diabetes, it is very important to figure out the amount of eaten food. The meal state can be checked by asking or directly looking at the meal plate. However, directly checking the meal state is not only inconvenient, but also has problems where the meal intake measurement is inaccurate and biased [1]. Various studies have been conducted on automated methods to objectively recognize food types and meal intake amounts. After object detection improves noticeably through artificial neural networks, various studies [2–5] have been conducted to detect foods in an image through object detection networks. Even though the performance of the food detection improves noticeably through the artificial neural networks, it is still difficult to measure the food volumes in the single image. The typical RGB image contains no 3D spatial information, so the 3D shape of the food is hard to reconstruct from the image. Several attempts [6–11] have been made to estimate the food volumes using stereo vision images or an RGB-D image. Some studies [12–17] estimate the food volume based on a shape-known reference object in a single image. Nevertheless, these are unsuitable for practical applications so far because the meal intake amount estimation is possible under limited conditions.

Institutions such as welfare centers analyze the dietary status of people who are receiving certain meal services to manage their health. The health management is achieved by checking the food types and measuring the meal intake amount. Since the meal intake amount is measured in some fixed levels, a slight imprecision of the meal amount measurement is acceptable. In this paper, we propose the methods of the food type classification and the intake amount estimation through an image pair for pre- and post-meal.

The flow of the proposed method is as follows: Mask R-CNN [18], which detects object regions in pixel units, finds the food regions and classifies the food types in the pre- and post-meal images. The post-meal image is corrected by homography transformation through

two meal plate regions in the images. For each food, the 3D shape type is determined as one of a spherical cap, a cone, and a cuboid based on the food type. The food volume is calculated through the 3D shape type and the food area. The meal intake amount is estimated as the difference in the food volumes in the pre- and post-meal images.

The contributions of this paper are that only pre- and post-meal images are needed to measure meal intake. That is, if a dedicated system attached with a camera takes a picture of a meal plate before and after a meal or takes a picture of a meal plate before and after a meal with a smartphone, then the food type can be automatically classified. In addition, the meal volume can be measured based on the detected food region and its predefined 3D shape without the need for a complex device to measure the volume.

## 2. Related Works for Food Detection and Meal Intake Amount Estimation

Research on the food type classification and the meal intake amount estimation is classified into methods by sensors and based on images. The food type and meal intake amount can be calculated by the sensors that measure sound, pressure, inertia, and so on, caused by eating food. The meal intake amount is measured by attaching weight or pressure sensors to a tray where meal plates are placed [19,20]. However, the estimation methods by the tray with the sensors have the disadvantage of extremely limited mobility. Some wearable devices can be utilized to measure the meal intake amount [21–23]. Olubanjo et al. [21] classifies the food type and measures the meal intake amount by template matching on the characteristics of sounds generated by chewing. Thomaz et al. [22] recognizes the food types and the meal intake amount through a smartwatch with a three-axis accelerometer. However, these methods cannot estimate the meal intake amount while the wearable device is not worn.

Recently, the technology for object detection in images has greatly been improved due to the development of artificial intelligence technology through deep neural networks. Various studies [2–5] for the food type classification have been conducted through object detection networks such as YOLO [24], VGG19 [25], AlexNet [26], and Inception V3 [27]. Wang et al. [28] pre-processes the meal image through morphological operators before detecting foods through an object detection network. The food detection methods through deep neural networks outperform the traditional methods [29], which are scale invariant feature transform (SIFT), histogram of oriented gradients (HOG), and local binary patterns (LBP). However, it is very difficult to measure the meal intake amount through only one image.

For the image-based methods, the meal intake amount should be measured by images captured from two or more viewpoints, an RGB-D image, or prior modeling of the foods. The methods based on multiple images [6,7] measure the food volume by reconstructing the 3D shape through correspondences among the pixels of images. Bándi et al. [6] finds some feature points in the food images captured with stereo vision in order to generate a point cloud that is a set of points on 3D space. The food volumes are measured through the point cloud for images. The food volume can also be estimated through the RGB-D image, which is added as a channel for the distance to subjects [8–11]. Lu et al. [8] detects the food regions and classifies the food types through convolution layers through RGB channels of the captured image. The food volumes are estimated by reconstructing 3D surfaces through a depth channel that has distance information. The 3D shape can be predicted by pre-modeling for the template of the food or bowl [30–32]. Hippocrate et al. [30] estimates the food volumes based on a known bowl shape. The food volume estimation through a single image requires a distinct reference object [12–14]. The meal intake amount can be estimated by the ratio of the number of pixels between the food and the reference object regions [12]. However, this estimation method has a large error for thin food. Smith et al. [13] calculate a projection model from a food object to the image through the reference object region. After that, one 3D shape type among sphere, ellipsoid, and cuboid is manually assigned to the food region to estimate the food volume. Liu et al. [14] crop a food region without background through Faster R-CNN, Grabcut algorithm, and median filter. The relative

food volume is estimated through the CNN network that learns the relationship between the food image without background and the food volume. The actual volume is calculated through the area ratio between the food and the size-known reference object. The estimation methods based on the reference object have an inconvenience where the reference object should be captured with food. In order to overcome this inconvenience, a shape-known dish or bowl can be utilized as the reference object [15–17]. Jia et al. [15] generate a 3D bowl model by measuring distances between line marks in the graduated tape attached at the bowl to estimate the meal intake amount. Fang et al. [16] and Yue et al. [17] calculate the camera parameters such as a camera pose and a camera focus length through the shape-known dish. The 3D food shape is generated through the camera parameters to calculate the food volume. However, these methods can only estimate the foods on certain containers.

## 3. Food Classification and Meal Intake Amount Estimation through Deep Learning

The proposed method classifies food types and estimates meal intake amounts by comparing pre- and post-meal images. Figure 1 shows the flow of the proposed method. In both pre- and post- meal images, the regions of the food and the meal plates are detected and the food types are classified through the object detection network. In order to compare food amounts between two images under the same capturing environment, the post-meal image is corrected by homography transformation through the detected plate regions in the both images. For each food in the images, the 3D shape type is determined as one of a spherical cap, a cone, and a cuboid based on the food type. The food volume is estimated through the 3D shape types and the food area. The meal intake amount is estimated by comparing the food volumes between the pre- and post-meal images.

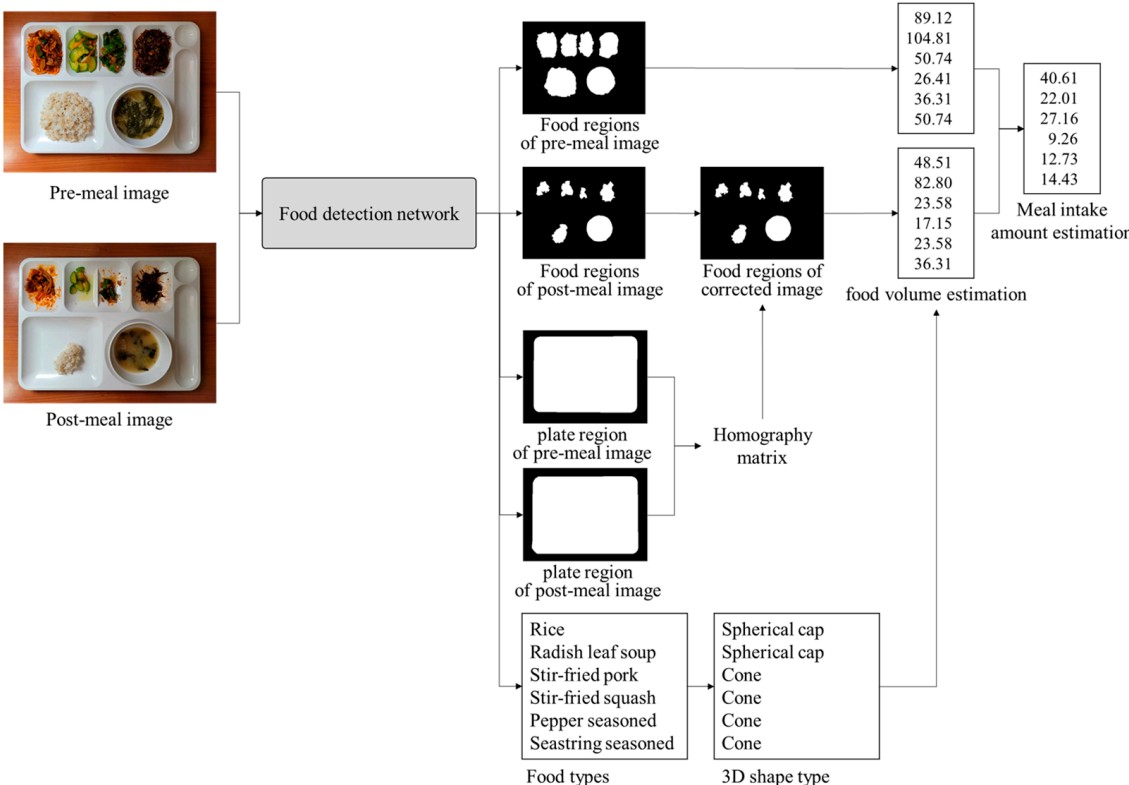

**Figure 1.** Flow of the proposed method.

### 3.1. Dataset

Images with Korean food and the meal plate which are captured by ourselves are utilized as the dataset for the proposed method. The dataset has 20 types of Korean food as

shown in Table 1. The foods in the dataset are classified into three categories as follows: rice, soup, and side-dish. The dataset has 520 images. The dataset is divided into 416 training images and 104 validation images at a ratio of about 8:2. The foods are placed on the concave surface within the designated meal plate as shown in Figure 2. The size of the meal plate is 40.5 cm × 29.5 cm. The soup food is served in a separate bowl whose radius is 3.5 cm.

**Table 1.** Categorization of foods in dataset.

| Category | Food Name | No. of Images in Training Set | No. of Images in Validation Set |
|---|---|---|---|
| Rice | Rice | 412 | 95 |
| Soup | Bean sprout soup | 108 | 16 |
| | Miso soup | 216 | 32 |
| | Radish leaf soup | 80 | 12 |
| | Seaweed soup | 68 | 12 |
| Side-dish | Eggplant | 86 | 13 |
| | Fruit salad | 43 | 5 |
| | Grilled fish | 246 | 43 |
| | Jeon | 104 | 23 |
| | Kimchi | 258 | 29 |
| | Pepper seasoned | 81 | 11 |
| | Seastring seasoned | 84 | 14 |
| | Stewed fish | 185 | 20 |
| | Stir-fried fish cake | 84 | 23 |
| | Stir-fried mushroom | 121 | 29 |
| | Stir-fried octopus | 58 | 5 |
| | Stir-fried pork | 179 | 29 |
| | Stir-fried squash | 87 | 8 |
| | Tofu | 116 | 26 |
| | Yellow pickled radish | 142 | 17 |

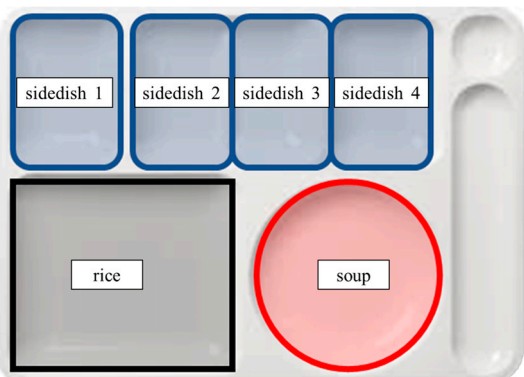

**Figure 2.** Food placement in meal plate.

Data augmentation is applied in order to increase the efficiency of the network training. Data augmentation is a strategy to increase the number of the data for the network training. Data augmentation increases the data without disappearance of main characteristics through image processing. In the proposed method, the image burring, the image rotation, and the image flip are applied probabilistically for the data augmentation as shown in Figure 3.

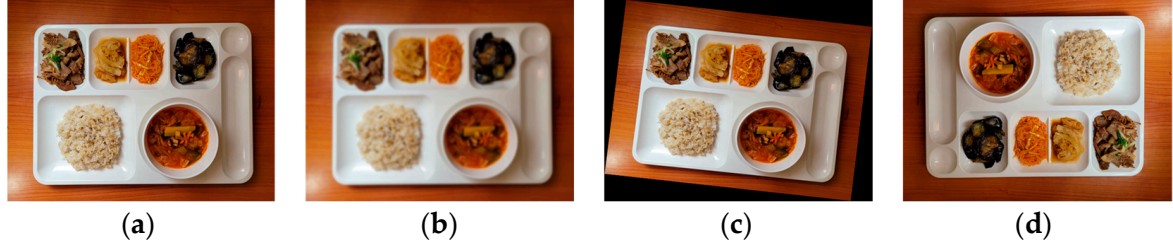

**Figure 3.** Data augmentation for network training: (**a**) original image; (**b**) blurring; (**c**) rotation; (**d**) flip.

### 3.2. Food Detection through Mask R-CNN

The bounding boxes detected by the usual object detection are inappropriate for estimating the food volume since it does not have information on the shapes of the foods. The food volume estimation requires the food region in pixel units. The proposed method detects the foods through Mask R-CNN [18] which is possible for the object detection in pixel units. ResNet-50 [33] is applied as the backbone of Mask R-CNN. Figure 4 shows the flow for food detection through Mask R-CNN. ResNet-50 extracts the feature maps from a meal plate image. The regions of interest (ROIs) are extracted by a region proposal network (RPN) from the feature maps. ROIAlign crops and interpolates the feature maps through ROIs. For each ROI, the food type and the bounding box are predicted through fully connected layers (FC layers). The food region in pixel units for each ROI is predicted through a fully convolutional network (FCN) [34]. Figure 5 shows the results of the food type classification and the food region detection through Mask R-CNN.

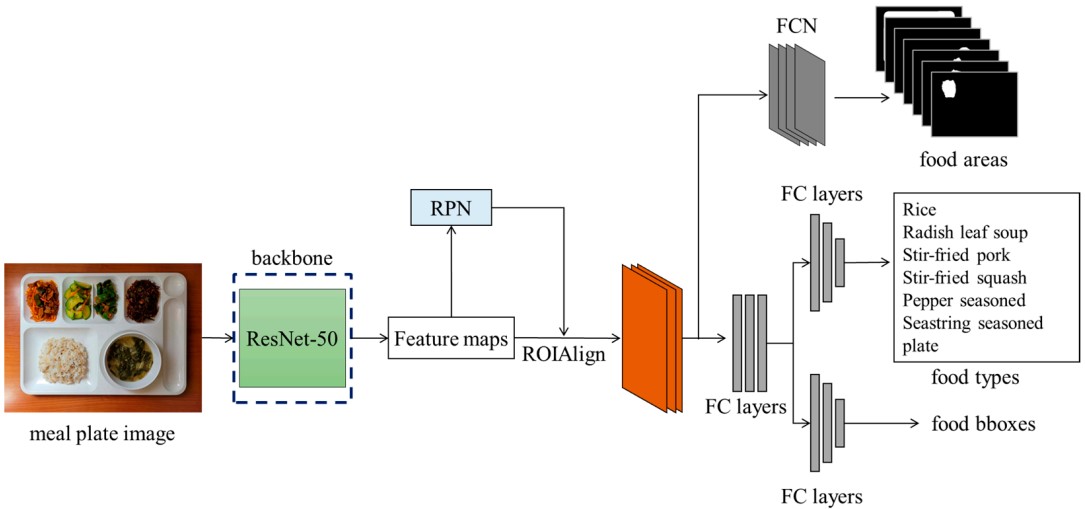

**Figure 4.** Food detection through Mask R-CNN.

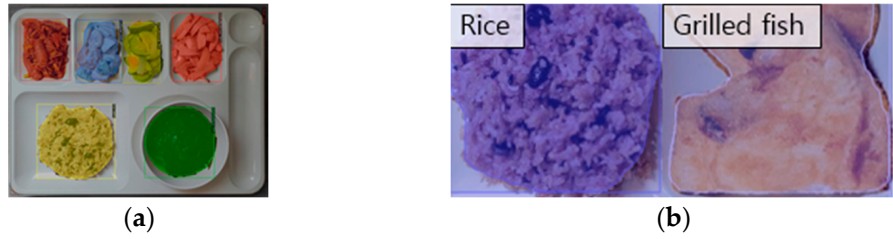

**Figure 5.** Food detection results: (**a**) food region detection; (**b**) food type classification.

### 3.3. Image Correction for Food Amount Comparison

The size of an object in an image depends on the capturing environments such as the camera pose and the distance. Therefore, both pre- and post-meal images should have the same capturing environment for accurately comparing the food amounts. However, the capturing environments of two meal images are often different in usual situations. In order to match both images to the same capturing environment, the post-meal image is corrected based on the meal plate regions in the both images. For each image, one rectangle and its vertices are found that enclose the meal plate region through rotating calipers algorithm [35]. A homography matrix is calculated from a pair of four vertices in two images. The post-meal image is corrected by the homography transformation with the calculated matrix as shown in Figure 6.

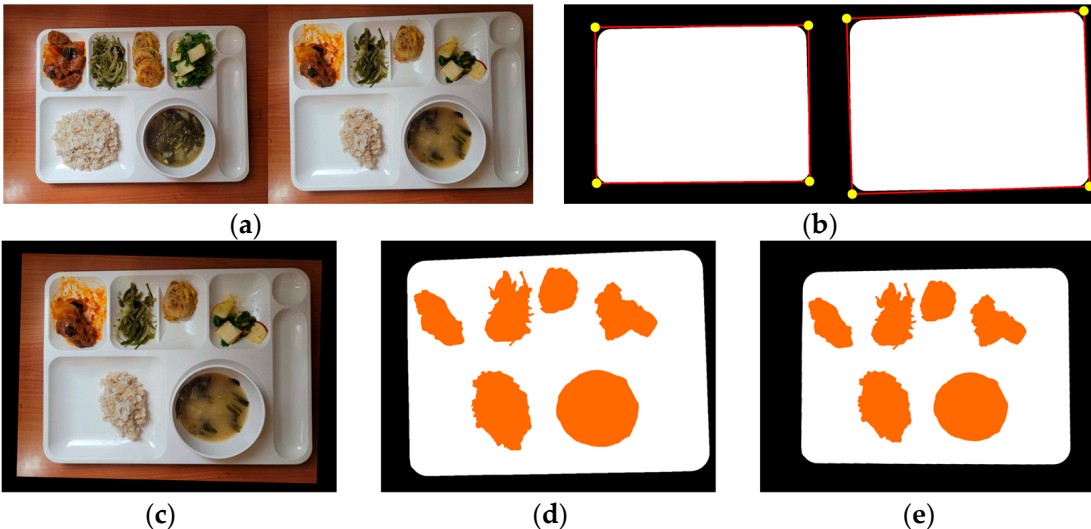

**Figure 6.** Image correction to have same capturing environment: (**a**) pre- and post-meal images; (**b**) rectangles and vertices enclosing meal plate regions; (**c**) corrected post-meal image by homography transformation; (**d**) food regions of post-meal image before correction; (**e**) food regions after correction.

### 3.4. Meal Intake Amount Estimation

The meal intake amounts are estimated as the differences in the food volumes between the pre- and post-meal images. Estimating a volume from an image is known to be a very challenging task. Nevertheless, we propose a method of the food volume estimation by assuming that the foods on the meal plate have a few simple 3D shape types. In the proposed method, the food is modeled in a 3D shape according to the type and the area as shown in Figure 7.

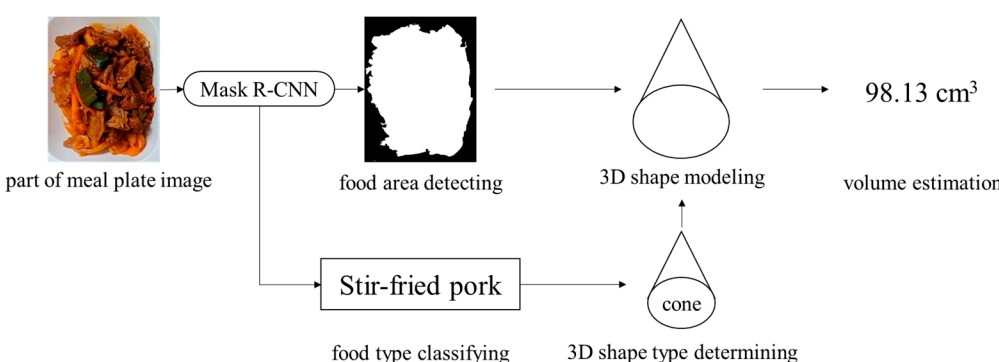

**Figure 7.** Flow of food volume estimation.

Food has a specific 3D shape depending on its food type. Foods in the rice and the soup categories are similar to a spherical cap shape as shown in Figure 8a. In Figure 8b, the foods which consist of bunches of an item are shaped like cones. Figure 8c shows that the shape of the food is close to cuboids if the food is one chunk. The proposed method determines the 3D shape types for the characteristic of food as a spherical cap, a cone, or a cuboid shape. Table 2 shows 3D shape types by the food types.

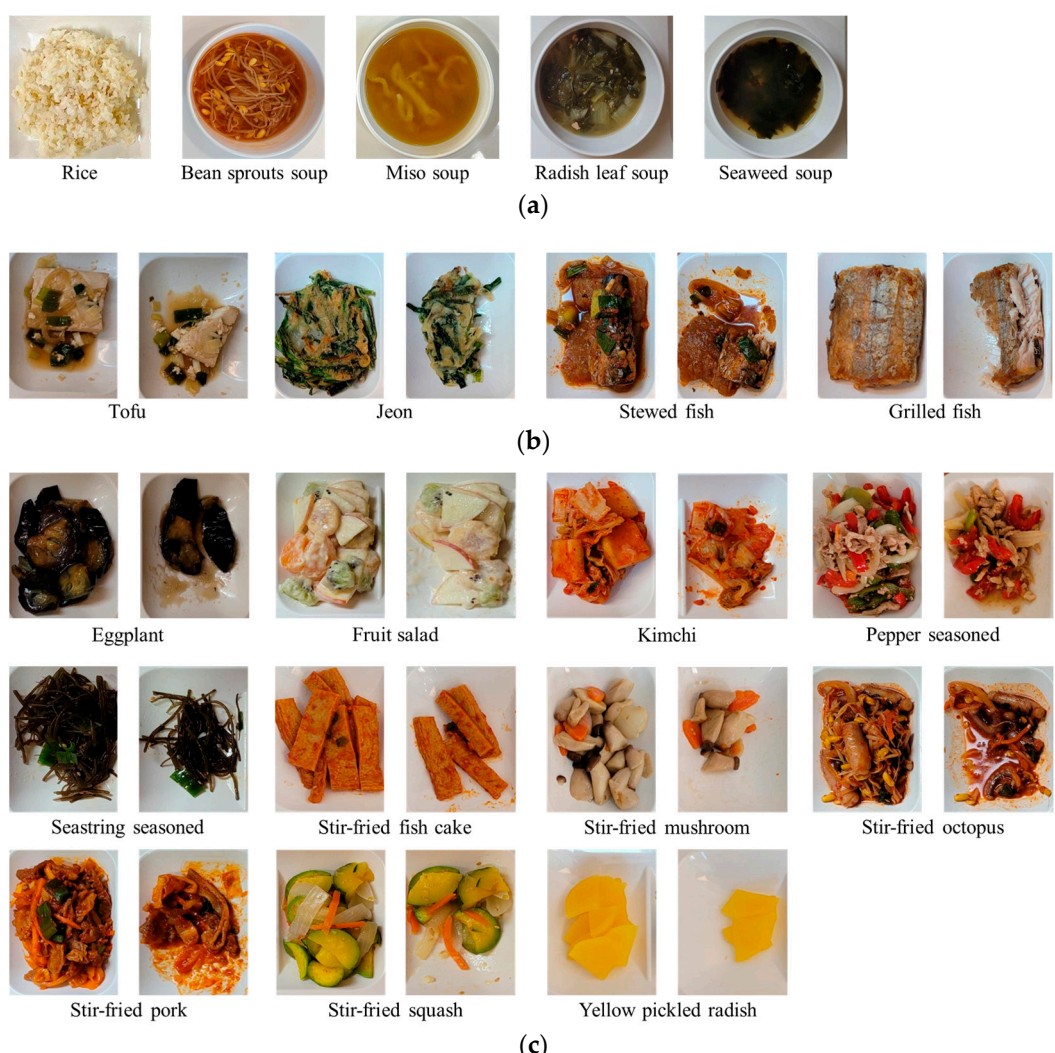

**Figure 8.** Shapes of foods: (**a**) spherical cap shape type; (**b**) cuboid shape type; (**c**) cone shape type.

**Table 2.** 3D shape types for food types in proposed method.

| Category | Food Name | 3D Shape Types | Category | Food Name | 3D Shape Types |
|---|---|---|---|---|---|
| Rice | Rice | spherical cap | | Fruit salad | Cone |
| Soup | Bean sprout soup | spherical cap | Side-dish | Kimchi | cone |
| | Miso soup | spherical cap | | Pepper seasoned | cone |
| | Radish leaf soup | spherical cap | | Seastring seasoned | cone |
| | Seaweed soup | spherical cap | | Stir-fried fish cake | cone |
| Side-dish | Tofu | cuboid | | Stir-fried mushroom | cone |
| | Jeon | cuboid | | Stir-fried octopus | cone |
| | Stewed fish | cuboid | | Stir-fried pork | cone |
| | Grilled fish | cuboid | | Stir-fried squash | cone |
| | Eggplant | cone | | Yellow pickled radish | cone |

The food volume is estimated through the base area and the 3D shape types. The base area is calculated through the actual meal plate area as follows:

$$A_{food} = \frac{n(p)}{n(f)} \times A_{plate},\tag{1}$$

where $n(.)$ is the number of pixels; $p$ and $f$ are the regions of the meal plate and the food, respectively; and $A_{food}$ and $A_{plate}$ are the base area and the meal plate area, respectively.

Even though the base area and the shape type are determined, the 3D shape can be different. For example, the shape of a cone with a specific base depends on its height. It is very difficult to estimate the height through one image. However, the foods cannot be sloped beyond a certain angle, which is the angle of repose. Though the angles of repose are different depending on the material of the food, the proposed method supposes that the angle of repose is 30 degree which is the average angle [36]. In other words, the food is assumed to have the slope angle of 30 degree.

The food volume is estimated through the 3D shape type of the food. For the food of the spherical cap type, the food height $h$ is equal to $R - a$ as shown in Figure 9a, where $R$ is the radius of the whole sphere, and $a$ is a distance between the spherical center and the base area. The slope angle $\theta_{slope}$ is 30 degree; thus,

$$\sqrt{\left(\frac{R}{a}\right)^2 - 1} = \tan\theta_{slope} = 1/\sqrt{3},\tag{2}$$

From (2), $a$ and $h$ are

$$a = \sqrt{3}R/2,\tag{3}$$

$$h = R - a = \left(2 - \sqrt{3}\right)R/2.\tag{4}$$

The base area $A_{food}$ is equal to $\pi r^2$; then, $r$ is

$$r = \sqrt{A_{food}/\pi}.\tag{5}$$

Since the angle between $R$ and $h$ is also equal to $\theta_{slope}$, $R$ is calculated as follows:

$$R = r \times \csc\theta_{slope} = 2r = 2\sqrt{A_{food}/\pi}.\tag{6}$$

The result of substituting $R$ into (4) is

$$h = 2\sqrt{A_{food}/\pi} \times \left(2 - \sqrt{3}\right)/2 = \left(2 - \sqrt{3}\right)\sqrt{A_{food}/\pi}.\tag{7}$$

The volume equation of the spherical cap is

$$V_{sph} = \frac{1}{3}\pi h^2(3R - h).\tag{8}$$

By substituting (6) and (7) into (8), the food volume $V_{sph}$ for the spherical cap is calculated as follows:

$$V_{sph} = \frac{16 - 9\sqrt{3}}{3\sqrt{\pi}}\left(A_{food}\right)^{3/2} \approx 0.08\left(A_{food}\right)^{3/2}.\tag{9}$$

Though the food in the soup category has the spherical cap shape, it is liquid contained in a bowl. The slope angle of the food is not fixed at 30 degree but depends on the curvature of the bowl. If $R$ is given, $h$ is calculated through the Pythagorean theorem as follows:

$$h = R - \sqrt{R^2 - r^2}. \tag{10}$$

For the food with the cone shape, the radius of the base surface $r$ is also estimated through (5). The food height $h$ is calculated to be $r \tan \theta_{slope} = \sqrt{A_{food}/3\pi}$ as shown in Figure 9b. The food volume of the cone-shaped food $V_{cone}$ is estimated as follows:

$$V_{cone} = \frac{1}{3} A_{food} h \approx 0.11 \left( A_{food} \right)^{3/2}. \tag{11}$$

Though the height of the cuboid is hard to estimate, it can be empirically predicted that the food volume decreases in proportion to the base area of the food. Therefore, the food volume for the cuboid shape is

$$V_{cuboid} = A_{food} H_{food}, \tag{12}$$

where $H_{food}$ is the predefined height depending on the food type. For each food, the meal intake amount is estimated by comparing the food volumes between the pre- and post-meal images.

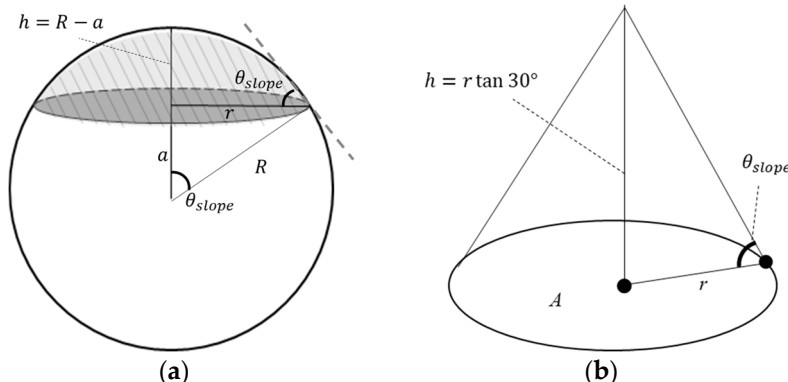

**Figure 9.** 3D food model for proposed method: (**a**) spherical cap; (**b**) cone.

## 4. Simulation

### 4.1. Simulation Results

We measure the accuracies of the food type classification and the meal intake amount by the trained Mask R-CNN. In the training of Mask R-CNN, the batch size is set as 64 and the epochs are set as 10,000, 30,000, 50,000, and 70,000. The performances of the proposed method are measured through 65 images with 206 food objects. In addition, Mask R-CNN and YOLOv8 [37] are applied as the food detection network to compare the performances of the proposed method.

The accuracies of the food existence detection and the food type classification are measured as shown in Table 3. The accuracy of the food existence increases as the epochs increase up to 50,000. All foods are detected with Mask R-CNN trained over 50,000 epochs. The accuracy of the food type classification is improved up to 97.57% until the food detection network is trained with 50,000 epochs. The accuracy of the food type classification hardly increases from the epochs greater than 50,000. Mask R-CNN is better than YOLOv8 for the food existence detection and the food type classification.

**Table 3.** Performance of food type classification for 206 foods.

| Network | Epochs | No. of Detected Objects (Accuracy %) | No. of Correct Classification (Accuracy %) |
| --- | --- | --- | --- |
| Mask R-CNN | 10,000 | 202 (98.06%) | 191 (92.72%) |
|  | 30,000 | 204 (99.03%) | 196 (95.15%) |
|  | 50,000 | 206 (100%) | 201 (97.57%) |
|  | 70,000 | 206 (100%) | 201 (97.57%) |
| YOLOv8 | 10,000 | 191 (92.72%) | 183 (88.83%) |
|  | 30,000 | 196 (95.15%) | 190 (92.23%) |
|  | 50,000 | 200 (97.09%) | 196 (95.15%) |
|  | 70,000 | 202 (98.06%) | 197 (95.63%) |

Table 4 shows the accuracies of the food type classification by the food categories through Mask R-CNN trained with 50,000 epochs. All foods in the rice category are accurately classified. However, some of the foods in the soup and the side-dishes categories are misclassified as different food types within the same category. The food detection network occasionally classifies foods as different food types with similar color.

**Table 4.** Performance of food type classification by food categories.

| Food Category | No. of Total Objects | No. of Detected Objects (Accuracy %) | No. of Correct Classification (Accuracy %) |
| --- | --- | --- | --- |
| Rice | 63 | 63 (100%) | 63 (100%) |
| Soup | 18 | 18 (100%) | 17 (94.44%) |
| Side-dish | 94 | 94 (100%) | 89 (94.68%) |

The accuracies of the food region detection are measured by calculating intersection of union (*IoU*) as follows:

$$IoU = \frac{n(g \cap d)}{n(g) + n(d) - n(g \cap d)} \tag{13}$$

where *g* and *d* are the ground truth and the detected regions, respectively. Figure 10 shows the accuracies of the food area detection. Similar to the food type classification, the accuracy of the food area detection is also increased up to 93.6% until the food detection network is trained with 50,000 epochs. Mask R-CNN is also more accurate than YOLOv8 for the food area detection.

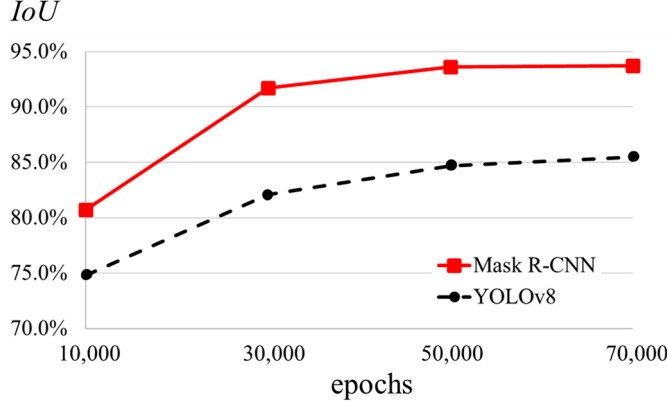

**Figure 10.** Accuracies of food area detection with comparison between Mask R-CNN and YOLOv8.

Figure 11 shows the images of the meal plates with average meal intakes of about 40%, 80%, and 90% for the food, respectively. The meal intake amounts for Figure 11



are estimated through the proposed method as shown in Table 5. The proposed method estimates the meal intake amounts closely to the actual amounts for most foods except the soup category. For foods in the soup category, the estimated intake amounts are smaller than the actual. The soup bowl has a small curvature, that is, a large slope. Therefore, the change in the food area is excessive small compared to the decrease in the food volume.

*4.2. Discussion*

The proposed method can accurately detect food regions in the meal image and classify food types. In addition, the meal intake amount can be estimated through a pair of images of the pre- and post-meal when foods are on the designated meal plate. RGB-D images or images from multiple viewpoints are not necessary to estimate the food volumes in the proposed method. However, the proposed method does not assume different foods mixed in a single dish as shown in Figure 12. The proposed method is not suitable for estimating precise food volume estimation with an extremely small tolerance.

Even though Korean foods are only covered, the proposed method can be applied to foods from other countries. The type detection for another nation foods is possible by training the object detection network with images of the corresponding foods. The 3D shape types for another nation's foods are similar to the types presented in this paper as shown in Figure 13. Therefore, this meal intake amount estimation can be widely applied for foods of various countries.

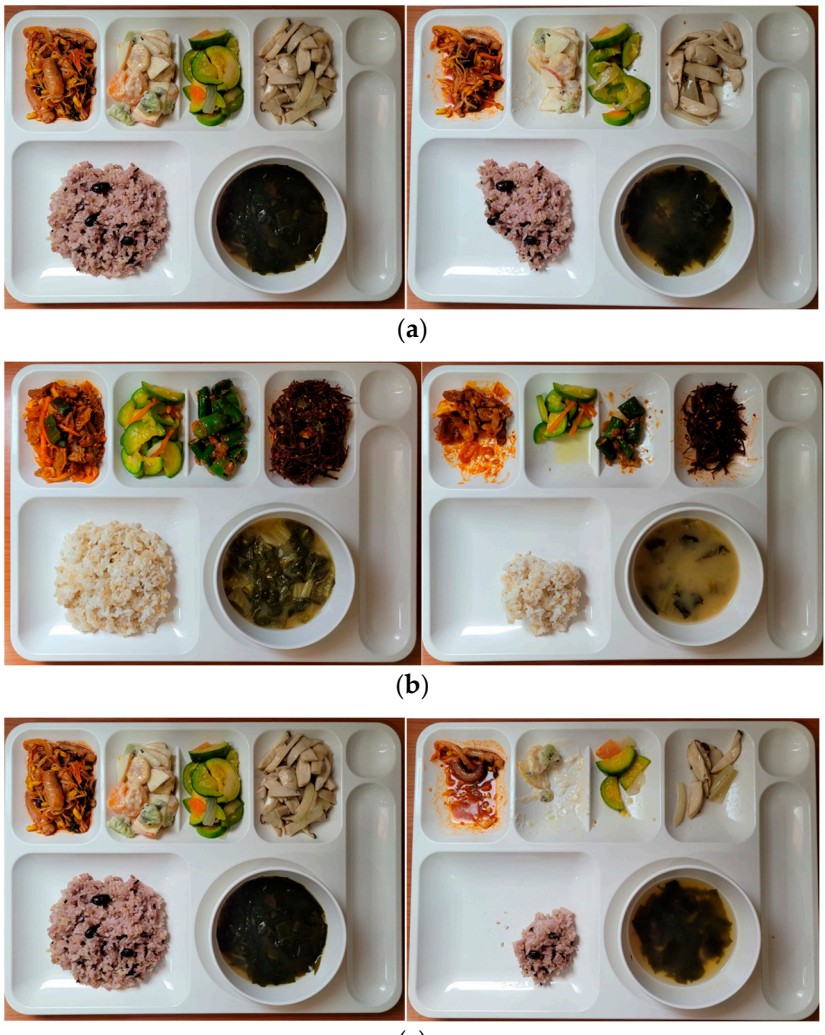

**Figure 11.** Pairs of images between pre- and post-meal: (**a**) intake of about 40%; (**b**) intake of about 80%; (**c**) intake of about 90%.

**Table 5.** Results of food volume estimation through proposed method.

| Target | Food Category | Food Name | Food Volume of Pre-Meal Image (cm$^3$) | Food Volume of Post-Meal Image (cm$^3$) | Meal Intake Amount (cm$^3$) |
|---|---|---|---|---|---|
| Figure 11a | Rice | Rice | 89.12 | 48.51 | 40.61 |
| | Soup | Seaweed soup | 104.81 | 82.80 | 22.01 |
| | Side-dish 1 | Stir-fried octopus | 50.74 | 23.58 | 27.16 |
| | Side-dish 2 | Fruit salad | 26.41 | 17.15 | 9.26 |
| | Side-dish 3 | Stir-fried squash | 36.31 | 23.58 | 12.73 |
| | Side-dish 4 | Stir-fried mushroom | 50.74 | 36.31 | 14.43 |
| Figure 11b | Rice | Rice | 89.12 | 26.41 | 62.71 |
| | Soup | Sirak soup | 104.81 | 49.69 | 55.12 |
| | Side-dish 1 | Stir-fried pork | 50.74 | 5.69 | 45.05 |
| | Side-dish 2 | Stir-fried squash | 36.31 | 5.69 | 30.62 |
| | Side-dish 3 | Pepper seasoned | 17.15 | 3.30 | 13.85 |
| | Side-dish 4 | Seastring seasoned | 36.90 | 17.15 | 19.75 |
| Figure 11c | Rice | Rice | 89.12 | 17.15 | 71.97 |
| | Soup | Seaweed soup | 104.81 | 37.73 | 67.08 |
| | Side-dish 1 | Stir-fried octopus | 50.74 | 4.54 | 46.20 |
| | Side-dish 2 | Fruit salad | 26.41 | 3.30 | 23.11 |
| | Side-dish 3 | Stir-fried squash | 36.31 | 4.54 | 31.77 |
| | Side-dish 4 | Stir-fried mushroom | 50.74 | 12.84 | 37.90 |

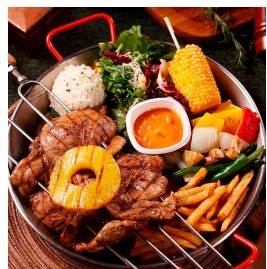

**Figure 12.** Various foods mixed in a single dish.

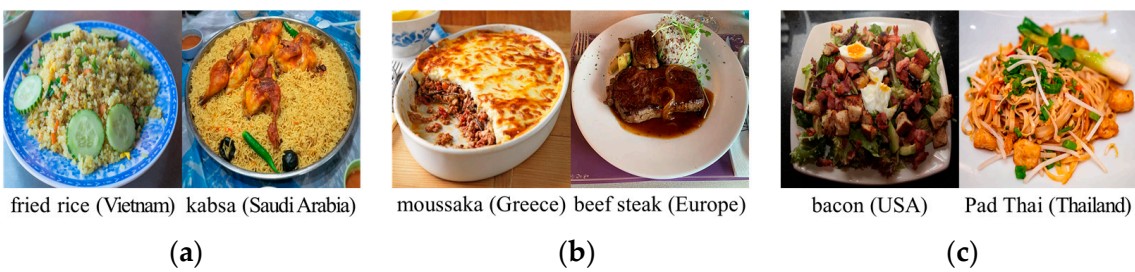

fried rice (Vietnam)  kabsa (Saudi Arabia)

moussaka (Greece)  beef steak (Europe)

bacon (USA)  Pad Thai (Thailand)

(**a**)  (**b**)  (**c**)

**Figure 13.** 3D shape types for foods of various countries: (**a**) spherical cap shape; (**b**) cuboid shape; (**c**) cone shape.

## 5. Conclusions

In this paper, we proposed the methods of the food type classification and the meal intake amount estimation. The food regions and the food types were detected through the Mask R-CNN. The post-meal image was corrected to have the same capturing environment based on the vertex points of the meal plate in the two images. The 3D shape type was determined as one of a spherical cap, a cone, and a cuboid for each food in the images. The food volumes were estimated through the detected area sizes and the 3D shape types. The meal intake amounts were estimated as the food volume differences between pre- and post-meal. In the simulation results, the accuracies of the food type classification and the

food region detection were up to 97.57% and 93.6%, respectively. The proposed method can be applied not only to Korean food, but also to other countries' foods, such as other Asian countries or countries in Europe. It is possible to analyze the ingested nutrients through the proposed method. The ingested nutrients are identified through the classified food types. The amounts of the ingested nutrients are calculated through the food types and the estimated meal intake amount. The nutrient analysis through the proposed method allows us to suggest a diet that provides a balanced nutrient intake. The adherence of a patient to dietary restrictions can be checked by analyzing the ingested nutrients. Moreover, it is possible to recommend the intake of the corresponding food for the part lacking in a specific nutrient.

**Author Contributions:** Conceptualization, J.-h.K., D.-s.L. and S.-k.K.; software, J.-h.K.; writing—original draft preparation, J.-h.K., D.-s.L. and S.-k.K.; supervision, S.-k.K. All authors have read and agreed to the published version of the manuscript.

**Funding:** This research was supported by the BB21+ Project in 2022 and supported by the MSIT (Ministry of Science and ICT), Korea, under the Grand Information Technology Research Center support program (IITP-2023-2020-0-01791) supervised by the IITP (Institute for Information & communications Technology Planning & Evaluation).

**Institutional Review Board Statement:** Not applicable.

**Informed Consent Statement:** Not applicable.

**Data Availability Statement:** Not applicable.

**Conflicts of Interest:** The authors declare no conflict of interest.

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
