# Peer review of "Food Classification and Meal Intake Amount Estimation through Deep Learning"

_applsci, doi:10.3390/app13095742_

Round 1

Reviewer 1 Report

The author claims that they propose a method to classify food types and estimate meal intake amounts in images through a deep-learning object detection network. Although the work seems meaningful, the models and methods used are relatively old and there are no innovative points in the article.

1>Please provide a detailed list of the contributions and innovations of this article.

2>It seems that there is no comparative experiment provided in the article. Is the method proposed in this article the pioneering article in this series?

3>The resolution of the images in this article is too low, please modify it.

4>Why does p appear in the explanation of Formula 13?

It is noted that your manuscript needs careful editing by someone with expertise in English editing paying attention to grammar, spelling, and sentence structure.

Author Response

We would like to express our appreciations for your review. We believe that your detailed comments and suggestions have contributed substantially to improve the presentation of our study, as well as its overall quality and the manuscript. Following, we offer replies to the points the reviewer addressed regarding the original manuscript.

Point 1: Please provide a detailed list of the contributions and innovations of this article.

Response 1: We added the contributions of the proposed method in Introduction section as follows.

  • The contributions of this paper are that only pre- and post-meal images are needed to measure meal intake. That is, if a dedicated system attached with a camera takes a picture of a meal plate before and after a meal or takes a picture of a meal plate before and after a meal with a smartphone, then the food type can be automatically classified. In addition, the meal volume can be measured based on the detected food region and its predefined 3D shape without the need for a complex device to measure the volume.

Point 2: It seems that there is no comparative experiment provided in the article. Is the method proposed in this article the pioneering article in this series?

Response 2: Most estimation methods for the meal intake amount have used multiple images or RGB-D images. Even though some studies have estimated the amount from a single image, the condition of the amount estimation was quite restricted. There are no prior articles that can be compared under the conditions assumed in this article. For aditoinally measuring the performances of the proposed method, YOLO family as another food detection network is compared with Mask R-CNN.

Point 3: The resolution of the images in this article is too low, please modify it.

Response 3: We enhanced the quality of the figures in the manuscript.

Point 4: Why does p appear in the explanation of Formula 13?

Response 4:  The ‘p’ is a typo. We corrected it to ‘d’

We would like to appreciate your detailed review again.

Reviewer 2 Report

1.      The English language used in the article has to be polished.

2.      The Introduction section of the paper, which covers the literature review,  is not well defined add more papers in literature review section.

3.      How are the hyper parameters used in the work chosen? Is it random or did the authors used any parameter tuning method?

4.       A more detailed analysis on the results obtained has to be added.

5.      How can the proposed work, enhanced meal intake amount estimation of the patients, explaine in detail.

6.      Figure 1. (Flow of food recognition) and figure 9 have very low quality, improve these figures.

7.      Authors cited Figure 12 a,b,c in table 6 but these figures are not given in manuscript.

Moderate editing of English language required before publishing the article.

Author Response

We would like to express our appreciations for your review. We believe that your detailed comments and suggestions have contributed substantially to improve the presentation of our study, as well as its overall quality and the manuscript. Following, we offer replies to the points the reviewer addressed regarding the original manuscript.

Point 1: The English language used in the article has to be polished.

Response 1: We polished the English in the manuscript and corrected the typos.

Point 2: The Introduction section of the paper, which covers the literature review, is not well defined add more papers in literature review section.

 Response 2: We added a new section for literature review about the food detection and meal intake amount estimation as follows.

  • Related Works for Food Detection and Meal Intake Amount Estimation

The researches of food type classification and meal intake amount estimation are classified into the methods by sensors or based on images. The type and intake amount of the food can be calculated by the sensors that measure sound, pressure, inertia, and so on caused by eating foods. The amount of food is measured by attaching weight or pressure sensors to the tray on which foods are placed [13-14]. However, the estimation methods by the tray with the sensors have disadvantage of extremely limited mobility. Some wearable devices can be utilized to measure the meal intake amount [15-17]. Olubanjo et al. [15] classifies the food type and measures the meal intake amount by template matching on the characteristics of the sound generated by chewing. Thomaz et al. [16] recognizes the food types and the meal intake amount through a smartwatch with a 3-axis accelerometer. However, these methods cannot estimate the meal intake amount while the wearable device is not worn.

Recently, the technology for object detection in images has greatly been improved due to the development of artificial intelligence technology through deep neural networks. Various studies [2-5] for the food type classification have been conducted through object detection networks such as YOLO [18], VGG19 [19], AlexNet [20], and Inception V3 [21]. Wang et al. [22] pre-processes the meal image through morphological operators before detecting foods through object detection network. The food detection methods through deep neural networks outperform the traditional methods, which are scale invariant feature transform (SIFT), histogram of oriented gradients (HOG), and local binary patterns (LBP) [23]. However, it is very difficult to measure the meal intake amount through only one image.

For the image-based methods, the meal intake amount should be measured by images captured from two or more viewpoints, a RGB-D image, or prior modeling of the foods. The methods based on multiple images [6-7] measure the food volume by reconstructing 3D shape through correspondences among the pixels of images. Bándi et al. [6] finds some feature points in the food images captured with stereo vision in order to generate a point cloud that is a set of points on 3D space. The food volumes are measured through the point cloud for images and the reference object whose real volume is known. The food volume can also be estimated through the RGB-D image, which is added as a channel for the distance to subjects [8-11]. Lu et al. [8] detects the food regions and classifies the food types through convolution layers through RGB channels of the captured image. The food volumes are estimated by reconstructing 3D surfaces through a depth channel that has distance information. The 3D shape can be predicted by pre-modeling for the template of the food or bowl [24-26]. Hippocrate et al. [24] estimates the food volumes based on a known bowl shape.

Point 3: How are the hyper parameters used in the work chosen? Is it random or did the authors used any parameter tuning method?

Response 3: We checked the performance by the number of the epochs in the network training. Since there was little performance improvement at 70000 epochs, the optimal epoch may be about 50000. 

Point 4: A more detailed analysis on the results obtained has to be added.

Response 4: We explained the detailed analysis of our simulation results.

  • Table 4 shows the accuracies of the food type classification by the food categories through Mask R-CNN trained with 50000 epochs. All foods in the rice category are accurately classified. However, some of the foods in the soup and side-dishes categories are misclassified as different food types within the same category. The food detection network occasionally classifies foods as different food types with similar color.
  • Fig. 11 shows the images of the meal plates with average meal intakes of about 40%, 80%, and 90% for the food, respectively. The meal intake amounts for Fig. 11 are estimated through the proposed method as shown in Table 5. The proposed method estimates the food intake amounts closely to the actual amounts for almost foods except the soup category. For foods in the soup category, the estimated intake amounts are smaller than the actual. The soup bowl has a small curvature, that is a large slope. Therefore, the change in the food area is excessive small compared to the decrease in the food volume.

Point 5: How can the proposed work, enhanced meal intake amount estimation of the patients, explain in detail.

 Response 5: We enhanced the explanation of the meal intake amount estimation of the patients as follows.

  • It is possible to analyze the ingested nutrients through the proposed method. The ingested nutrients are identified through the classified food types. The amounts of the ingested nutrients are calculated through the food types and the estimated meal intake amount. The nutrient analysis through the proposed method allows to suggest a diet that provides a balanced nutrient intake. The adherence of a patient to dietary restrictions can be checked by analyzing the ingested nutrients. Also, it is possible to recommend the intake of the corresponding food for the part lacking in a specific nutrient.

Point 6: Figure 1. (Flow of food recognition) and figure 9 have very low quality, improve these figures.

Response 6:  We improved the image quality of Figure 1 and 9.

Point 7: Authors cited Figure 12 a,b,c in table 6 but these figures are not given in manuscript.

Response 7:  We corrected wrong citations in Table 6.

We would like to appreciate your detailed review again.

Reviewer 3 Report

The idea of this paper sounds interesting but there are some comments need to be addressed:

1. Technical contributions and novelty of papers are missing. Please describe the significance of current study.

2. Figure 1 - Can be further enhanced by including the relevant figures to better illustrate the overall workflow. 

3. Table 1 - Any specific ratios used to split the datasets into training and validation sets? Please clarify and explain the rationale of using this ratio for data splitting.

4. Section 2.2 - Some details of Mask R-CNN used to detect the food areas and classify food types should be described in this subsection. 

5. Why choose Mask R-CNN over YOLO family algorithm to detect the food areas and classify food types? Any performance comparison done to justify the selection of algorithm?

6. Section 2.4- Although authors have provided a lot equations to calculate the volume of foods, it is still not clearly explained how to estimate the food amounts to be consumed. 

7. Table 2 - Any specific logic used to assign the 3D shape of each food? Some are logical but some seems not. For instance, why Kimchi is assigned as cone shape? It will be good for the author to modify the Table 2 by providing graphic information of foods and the designated 3D shape.

8. Eq. (13) seems incomplete. What is the left hand side of equation?

9. Quality of Figure 9 is quite poor and too small. Please enlarge and improve quality.

10. Please discuss the limitation of proposed methods?

11. Is the proposed method also applicable to Western foods or other Asian foods? Any specific modifications need to be made to ensure the proposed method can be adapted into different  types of foods? Please have in-depth discussion about this issue.

English language quality is good and easy to understand. Not much typo and grammatical errors are observed.

Author Response

We would like to express our appreciations for your review. We believe that your detailed comments and suggestions have contributed substantially to improve the presentation of our study, as well as its overall quality and the manuscript. Following, we offer replies to the points the reviewer addressed regarding the original manuscript.

Point 1: Technical contributions and novelty of papers are missing. Please describe the significance of current study.

Response 1: We added the contributions of the proposed method in Introduction section as follows.

  • The contributions of this paper are that only pre- and post-meal images are needed to measure meal intake. That is, if a dedicated system attached with a camera takes a picture of a meal plate before and after a meal or takes a picture of a meal plate before and after a meal with a smartphone, then the food type can be automatically classified. In addition, the meal volume can be measured based on the detected food region and its predefined 3D shape without the need for a complex device to measure the volume.

Point 2: Figure 1 - Can be further enhanced by including the relevant figures to better illustrate the overall workflow.

Response 2: We enhanced Fig. 1 to illustrate the workflow of the proposed method in more detail.

Point 3: Table 1 - Any specific ratios used to split the datasets into training and validation sets? Please clarify and explain the rationale of using this ratio for data splitting.

Response 3: We added the explanation of data splitting as follows.

  • The dataset was created by capturing the Korean foods and the meal plate ourselves. The dataset has 520 images. The dataset is divided into 416 training images and 104 validation images at a ratio of about 8:2.

Point 4: Section 2.2 - Some details of Mask R-CNN used to detect the food areas and classify food types should be described in this subsection.

Response 4: We added the description of the food detection through Mask R-CNN.

  • Fig. 4 shows the flow for food detection through Mask R-CNN. ResNet-50 extracts the feature maps from a meal plate image. The regions of interest (ROIs) are extracted by a region proposal network (RPN) from the feature maps. ROIAlign crops and interpolates the feature maps through ROIs. For each ROI, the food type and the bounding box are predicted through fully connected layers (FC layers). The food region in pixel units for each ROI is predicted through a fully convolutional network (FCN) [28]. Fig. 5 shows the results of the food type classification and the food region detection through Mask R-CNN.

Point 5: Why choose Mask R-CNN over YOLO family algorithm to detect the food areas and classify food types? Any performance comparison done to justify the selection of algorithm?

Response 5: We conducted the comparative simulation with YOLOv8.

Point 6: Section 2.4- Although authors have provided a lot of equations to calculate the volume of foods, it is still not clearly explained how to estimate the food amounts to be consumed.

Response 6:  We enhanced the explanation of the meal intake amount estimation.

  • The meal intake amounts for the food is estimated as the differences in the food volumes between the pre- and post-meal images. Estimating a volume from an image is known to be a very challenging task. Nevertheless, we propose a method of the food volume estimation by assuming that the foods on the meal plate have a few simple 3D shape types. In the proposed method, the food is modeled in a 3D shape according to the type and the area.

  Food has a specific 3D shape depending on its food type. Foods in the rice and soup categories are similar to a spherical cap shape as shown in Fig. 8 (a). In Fig. 8 (b), the foods which consist of bunches of an item are shaped like cones. Fig. 8 (c) shows that the shape of the food is close to cuboids if the food is one chunk. The proposed method determines the 3D shape types for the characteristic of food as a spherical cap, a cone, or a cuboid shape. Table 2 shows 3D shape types by the food types.

Point 7: Table 2 - Any specific logic used to assign the 3D shape of each food? Some are logical but some seems not. For instance, why Kimchi is assigned as cone shape? It will be good for the author to modify the Table 2 by providing graphic information of foods and the designated 3D shape.

Response 7:  We added the figures of actual foods to provide graphic information.

Point 8: Eq. (13) seems incomplete. What is the left hand side of equation?

Response 8:  We corrected Equation 13.

Point 9: Quality of Figure 9 is quite poor and too small. Please enlarge and improve quality.

Response 9:  We improved the image quality of Figure 9.

Point 10: Please discuss the limitation of proposed methods?

Response 10:  We added a discussion of the limitation as follows.

  • The proposed method does not assume different foods mixed in a single dish as shown in Fig. 12. The proposed method is not suitable for estimating precise food volume estimation with an extremely small tolerance.

Point 11: Is the proposed method also applicable to Western foods or other Asian foods? Any specific modifications need to be made to ensure the proposed method can be adapted into different types of foods? Please have in-depth discussion about this issue.

Response 11:  We explained the applicability of the proposed method to foods in other countries in 4.2 Discussion section.

  • Even though Korean foods are only covered, the proposed method can be applied to foods from other countries. The type detection for another nation foods is possible by training the object detection network to images of the corresponding foods. The 3D shape types for another nation foods are similar to the types presented in this paper as shown in Fig. 12. Therefore, this meal intake amount estimation can be widley applied for foods of various countries.

We would like to appreciate your detailed review again.

Round 2

Reviewer 1 Report

The author provides a certain explanation, but I think adding a comparison between Mask RCNN and YOLO does not effectively highlight the innovative points of the article. But the author claims that:‘There are no prior articles that can be compared under the conditions assumed in this article. ’ So I hope the author's statement is a responsible expression after thorough more researches on the latest article.

Minor editing of the English language required

Author Response

We would like to express our appreciations for your second review. The invaluable comments for analysis of the latest articles about meal intake amount estimation have been helped to express the contribution of our article more clearly. We believe that the quality of the manuscript is further improved for your detailed comments and suggestions.

The food volume estimation through a single image requires a distinct reference object. The meal intake amount can be estimated by the ratio of the number of pixels between the food and the reference object regions [12]. However, this estimation method has a large error for thin food. Smith et al. [13] calculate a projection model from a food object to the image through the reference object region. After that, one 3D shape type among sphere, ellipsoid, and cuboid is manually assigned to the food region to estimate the food volume. Liu et al. [14] crop a food region without background through Faster R-CNN, Grabcut algorithm, and median filter. The relative food volume can be estimated through the CNN network that learns the relationship between the food image without background and the food volume. The actual volume is calculated through the area ratio between the food and the size-known reference object. The estimation methods based on the reference object have an inconvenience that the reference object should be captured with food. In order to overcome this inconvenience, a shape-known dish or bowl can be utilized as the reference object. Jia et al. [15] generate a 3D bowl model by measuring distances between line marks in the graduated tape attached at the bowl to estimate the meal intake amount. Fang et al. [16] and Yue et al. [17] calculate the camera parameters such as a camera pose and a camera focus length through the shape-known dish. The 3D food shape is generated through the camera parameters to calculate the food volume. However, these methods can only estimate the foods on certain containers.

We would like to appreciate your detailed review again.

Reviewer 2 Report

None

Fine, minor editing of English language required

Author Response

We corrected English grammatical errors in the manuscript.

Reviewer 3 Report

Authors have done excellent jobs to address all comments given in the earlier review process. I am happy to recommend the acceptance of this manuscript. Well done and all the best.

No major issues with the quality of English language. The manuscript is well written and easy to follow.

Author Response

(The authors gave the same response as above.)
